# An atopic dermatitis-like murine model by skin-brushed cockroach Per a 2 and oral tolerance induction by *Lactococcus lactis*-derived Per a 2

**Mey-Fann Lee[1], Nancy M. Wang[2], Yi-Hsing Chen[3], Chi-Sheng Wu[1], Ming-Hao Lee[1], Yu-Wen Chu[4,5]***

**1** Department of Medical Research, Taichung Veterans General Hospital, Taichung, Taiwan, **2** Department of Biology, National Changhua University of Education, Changhua, Taiwan, **3** Division of Allergy, Immunology, and Rheumatology, Taichung Veterans General Hospital, Taichung, Taiwan, **4** Department of Pharmacy, Taichung Veterans General Hospital, Taichung, Taiwan, **5** Department of Pharmacy, College of Pharmaceutical Sciences, National Yang Ming Chiao Tung University, Taipei, Taiwan

* yuwenchu@vghtc.gov.tw

**Data Availability Statement:** All relevant data are within the paper and its Supporting Information files.

## Abstract

Atopic dermatitis (AD) is a complex, chronic inflammatory skin disease. An estimated 57.5% of asthmatic patients and 50.7% of rhinitis patients are allergic to cockroaches in Taiwan. However, the role of cockroaches in the pathogenesis of AD is undetermined. Oral tolerance might be another strategy for protecting against AD and allergic inflammation by regulating T helper 2 (Th2) immune responses. Aim to examine the underlying immunologic mechanism, we developed an AD-like murine model by skin-brushing with cockroach Per a 2. We also investigated whether the systemic inflammation of AD in this murine model could be improved by specific tolerance to *Lactococcus lactis*-expressing Per a 2, which was administered orally. Repeated painting of Per a 2 without adjuvant to the skin of mice resulted in increased total IgE, Per a 2-specific IgE, and IgG1, but not IgG2a. In addition, epidermal thickening was significantly increased, there were more scratch episodes, and there were increases in total white blood cells (eosinophil, neutrophil, and lymphocyte) and Th2 cytokines (Interleukin (IL)-4, IL-5, IL-9, and IL-13) in a dose-dependent manner. The results revealed that oral administration of *L. lactis*-Per a 2 ameliorated Per a 2-induced scratch behavior and decreased the production of total IgE, Per a 2-specific IgE, and IgG1. Furthermore, *L. lactis*-Per a 2 treatment also suppressed inflammatory infiltration, expressions of thymic stromal lymphopoietin (TSLP) and IL-31 in skin lesions, and downregulated splenic IL-4 and IL-13 in Per a 2-induced AD mice. This study provides evidence supporting that repeated brushing of aeroallergens to the skin leads to atopic dermatitis phenotypes and oral allergen-specific immune tolerance can ameliorate AD-like symptoms and systemic inflammation and prevent progression of atopic march.

**Funding:** This study was supported by grants from Taichung Veterans General Hospital (TCVGH-1106101B and TCVGH-1107312C). The funders had no role in study design, data collection and analysis, decision to publish, or preparation of the manuscript.

**Competing interests:** The authors have declared that no competing interests exist.

## Introduction

Atopic dermatitis (AD) is a common chronic inflammatory skin disease with a prevalence of 10–30% in children and 2–10% in adults worldwide [1–4]. AD is characterized by intense itching, pruritus, xerosis, lichenification, and recurrent eczematous lesions [4–8]. The pathophysiology of AD involves many multiple interacting factors, including susceptibility genes, dysfunctional epidermal barrier, high transepidermal water loss (TEWL), environmental allergens sensitization, and immunological factors [4, 9, 10]. AD is generally the first allergic manifestation of atopic march, which is then followed in later stages by other allergic disorders, such as food allergy, allergic rhinitis (AR), and asthma [4, 11]. Atopic march describes the progression of allergic diseases, and it has been estimated that approximately one-third of AD patients subsequently manifest asthma, while two-thirds experience the development of allergic rhinitis [12–16].

AD is a multifactorial inflammatory skin disease and has been associated with IgE sensitivity to airborne-derived, food-derived, and microbial allergens [17, 18]. Aeroallergens possess a well-documented ability to provoke asthma and allergic rhinitis [19], and are increasingly recognized as triggers of AD [15]. American cockroach (*Periplaneta americana*) allergens are one of the major indoor allergens in Taiwan [20]. In Taiwan, cockroach allergies developed in 57.5% of asthmatic patients and 50.7% of rhinitis patients [21, 22]. In respiratory allergies, American cockroach allergen Per a 2 correlates with asthma severity [22]. However, the role of Per a 2 in the initiation and activation of AD through the skin is limited. The specific underlying mechanism and relationship between atopic dermatitis and cockroach allergens require further investigation.

Allergen avoidance is difficult to implement for allergic patients with long-term indoor allergens exposure. Currently, medications for AD include antihistamines, topical and systemic corticosteroids, immunosuppressants, and biological agents, which play an essential role in reducing inflammation and repairing skin lesions [4, 23, 24]. Importantly, many targeting-type 2-cytokine therapies (for example, IL-4, IL-13, IL-5, IL-31 inhibitors and anti-IgE monoclonal antibodies) currently have shown benefits in AD patients [25, 26]. Allergen-specific immunotherapy (ASIT) is an effective IgE-mediated allergic disease therapy, whose efficacy has been demonstrated in house dust mites [27–30]. However, systemic and local adverse effects were reported in clinical trials for allergen immunotherapy [31–33].

An alternative immunotherapy strategy, oral tolerance (OT) has been implicated in the immune unresponsiveness induced by oral administration of antigens prior to epicutaneous sensitization [34]. However, relatively few OT or ASIT studies have been performed on cockroach allergens. *Lactococcus lactis* (*L. lactis*) is widely used as a delivery vehicle and live vector for mucosal antibody responses [35, 36]. Its safety and persistent colonization in the mucosal surface make *L. lactis* a good therapeutic delivery vehicle. We previously developed a triple-aeroallergen vaccine for Der p 2 (mite), Per a 2 (roach), and Cla c 14 (mold) using *L. lactis* expression system, which was effective and safe in preventing airway allergy without adverse effects in a murine model [37]. Thus, we explore the potential of *L. lactis* as an OT vector for Per a 2-specific allergen for AD patients. A 6-week sensitization protocol of the AD murine model was developed by skin-painting with cockroach allergen Per a 2. The effectiveness of oral administration of *L. lactis*-Per a 2 was then evaluated.

OT of the cockroach allergen might be an option to develop as a potential therapeutic drug or preventive supplement for atopic dermatitis. These OT products may exert fewer side effects than conventional ASIT and are cheaper than biologicals for AD. Moreover, they offer considerable promise as the basis for novel preventive or therapeutic strategies for atopic dermatitis.

## Materials and methods

### Preparation of recombinant Per a 2 proteins for sensitization

As previously described expression of American cockroach allergen Per a 2 recombinant proteins was used vectors pET30 in *Escherichia coli* [22]. Plasmid pET30 contains a His-tag sequence, which is expressed as a stretch of six histidine residues at the N-terminal end of the target protein. Purification of *E.coli*-expressed Per a 2 protein (*E*-rPer a 2) was conducted by rapid affinity chromatography (Novagen, Darmstadt, Germany). The subsequent purification of the recombinant protein was performed by Endotoxin Detoxi-Gel (Pierce, Illinois, USA), followed by its sterilization used with a 0.22-μm syringe filter (Millipore, Billerica, MA, USA), as described previously [37].

### Experimental design of an atopic dermatitis-like murine model

Six-week-old female BALB/c mice were obtained from the National Laboratory Animal Center in Taiwan and used for the present experiments. The present study was implemented in strict accordance with the "Guideline for the Care and Use of Laboratory Animals" established by the Council of Agriculture in Taiwan. All animal experiments were rigorously peer-reviewed and approved by the Institutional Animal Care and Use Committee of Taichung Veterans General Hospital (approval number: La-1091738). The mice were sacrificed by inhaling $CO_2$ and the flow rate of $CO_2$ was displaced 30–70% of the cage volume per minute. We developed a new protocol with 6-week allergen sensitization that involves less animal distress and reduces their pain. All efforts were made to minimize suffering.

The present experiment involved an optimal dose-schedule for induction of AD-like inflammation and symptoms with cockroach allergen Per a 2 via skin-brushed in BALB/c mice (Fig 1). Briefly, abdominal fur was removed with depilatory cream, followed by exposure

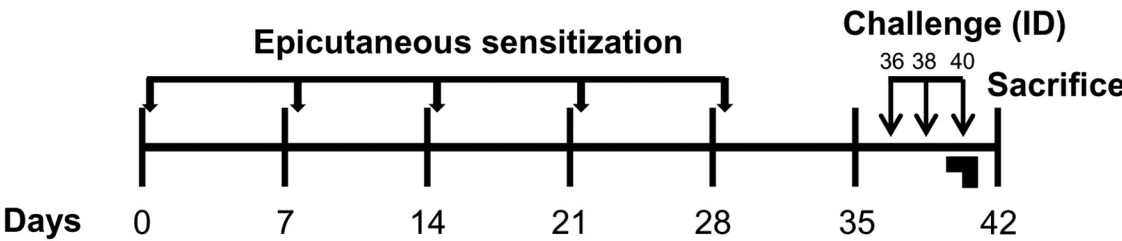

| | Group | N | Sensitization with abdominal skin painting to dry (100 μl) | Challenge (ID) (20 μl) |
|---|---|---|---|---|
| 1 | PBS | 5 | PBS | PBS |
| 2 | rPer a 2-10 | 5 | 10 μg of *E*-rPer a 2 | 1 μg of *E*-rPer a 2 |
| 3 | rPer a 2-50 | 5 | 50 μg of *E*-rPer a 2 | 5 μg of *E*-rPer a 2 |
| 4 | rPer a 2-100 | 5 | 100 μg of *E*-rPer a 2 | 10 μg of *E*-rPer a 2 |

**Fig 1. Dose finding schedule of cockroach allergen Per a 2-induced allergic dermatitis via skin-painting in BALB/c mice.** Purified proteins of recombinant Per a 2 were applied epicutaneously once per week for 5 weeks. To examine the response of sensitization, scratching behavior was videoed on day 40 after the intradermal (ID) challenge. Serum samples were obtained from the submandibular vein bi-weekly and preserved at -20°C until analysis. All mice were sacrificed on the 42nd day and the ID site of the skin and spleen were excised for further study.

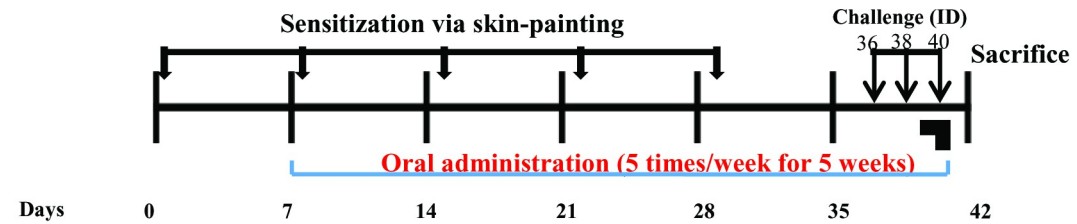

| | Group | N | Abdominal painting to dry (100 µl)<br>(Days 1, 7, 14, 21, 28) | Oral administration (200 µl)<br>(Weeks 2-6, 5 times/week for 5 weeks) | Challenge (ID, 20 µl) |
|---|---|---|---|---|---|
| 1 | T-sham | 6 | 50 µg of *E*-rPer a 2 | PBS | PBS |
| 2 | T-VO | 6 | 50 µg of *E*-rPer a 2 | NZ3900- 1x10⁹ CFU/day | 5 µg of *E*-rPer a 2 |
| 3 | T1-Per a 2 | 6 | 50 µg of *E*-rPer a 2 | NZ3900/pNZ8149-Per a 2,<br>5x10⁹ CFU/day (containing 1 µg of expressed protein) | 5 µg of *E*-rPer a 2 |
| 4 | T2-Per a 2 | 6 | 50 µg of *E*-rPer a 2 | NZ3900/pNZ8149-Per a 2<br>1x10¹⁰ CFU/day (containing 2 µg of expressed protein) | 5 µg of *E*-rPer a 2 |

**Note: Prior to oral, all mice were deprived food for 2 h, but not water.**

**Fig 2. Timeline and grouping of oral *L.lactis*-Per a 2 in a Per a 2-painting mouse model.** BALB/c mice were skin-painted *E*-rPer a 2 once per week for 5 weeks. On days 7–40, mice were intragastrically (IG) administered 200 µl of syrup containing 1 µg (group T1-Per a 2), 2 µg (group T2-Per a 2) of the Per a 2-expressed protein, NZ3900 *L.lactis* alone (group T-VO), or PBS (group T-sham) once a day on weekdays for 5 weeks. The scratching behavior was videoed on day 40 after the intradermal (ID) challenge. Serum samples were obtained from the submandibular vein bi-weekly and preserved at -20˚C until analysis. All mice were sacrificed on the 42nd day and the ID site of the skin and spleen were excised for further study.

to recombinant allergens in 100 µl of phosphate-buffered saline (PBS) or PBS alone spread on the abdominal skin to air dry. The size of the area on the shaved abdominal skin that was painted with sensitization solution measured about 1 cm$^2$. Mice were exposed once a week for a total of 5 exposures. From days 36 to day 40, the mice were challenged intradermally (ID) with three doses of allergens on three consecutive days. On the 40th day, immediately after the ID challenge, scratching behavior was videotaped. Serum samples from the submandibular vein were obtained bi-weekly and preserved at -20˚C until analysis. On the 42nd day, all mice were sacrificed, followed by the excision of their skin and spleen for subsequent investigation.

For the OT treatment, the established AD-like mice were orally administered with recombinant *L. lactis* Per a 2. The sensitization and treatment schedule is summarized in Fig 2. Mice were intragastrically (IG) administered 200 µl of syrup containing 1 µg (as group T1-Per a 2) or 2 µg (T2-Per a 2) of the expressed Per a 2 protein once a day on weekdays for a total of 25 times during days 7 to 40. Control mice (group T-VO) received *E*-rPer a 2 at the same time of painting and oral administration of NZ3900 *L.lactis* alone.

## Scratching behavior

After the last allergen challenge on the 40th day, the scratching behaviors of mice were videotaped for 1 hour. Counts of scratching around the ID sites were recorded and quantified through video playback, as described previously [38, 39].

## Histological analysis of skin lesions

The mice were euthanized 24 hours after the last allergen challenge and the skin lesions were excised immediately. The removed skin specimens were preserved and fixed in 10% neutral formalin and subsequently embedded in paraffin. Samples were sectioned at 5 µm thickness and subsequently stained with hematoxylin and eosin (H&E). All slides were observed under a Hamamastu NanoZoomer 2.0 HT slide scanner (Hamamastu, Japan) at 400-fold

magnification. As described previously, the quantification of infiltrating inflammatory cells of each mouse were conducted within H&E-stained skin sections by medical technicians [38].

## Measurement of total IgE

Based on the manufacturer's instructions, the total IgE levels were determined through an IgE mouse ELISA kit (Thermo Fisher Scientific). The 96-well plates (Nunc) were coated overnight at 4˚C with 5 μg/ml monoclonal antibody. Thereafter, all experiments were conducted at room temperature. After washing twice with PBS/0.05% Tween 20, the plates were blocked with PBS added 1% BSA for 2 hours and subsequently incubated for 2 hours with diluted serum (1:10) or 2-fold serial dilutions of mouse IgE standards. Following the washing step, the detection antibody (1:250) was added and incubated for 2 hours. Then plates were washed, and strepta-vidin-horseradish peroxidase (HRP) conjugate (1:400) was added and subsequently incubated for 30 minutes. To facilitate visualization, tetramethylbenzidine (TMB) was added and the reaction was terminated using 1 M phosphoric acid, followed by detection with an ELISA reader (TECAN, Austria) at 450 nm.

## Analysis of Per a 2-specific IgE, IgG1, and IgG2a antibodies by indirect ELISA

The microtiter plates (Maxisorp, Nunc) were coated with rPer a 2 (100 μl from 5 μg/ml in 100 mM $NaHCO_3$, pH9.6). Following blocking with 2% BSA, sera were diluted at 1:1000 for IgG1, 1:100 for IgG2a, and 1:10 for IgE, and afterward incubated overnight at 4˚C. To analyze IgE level, the plates were incubated with primary biotin-conjugated rat anti-mouse IgE (1:4000) and followed by secondary peroxidase-conjugated streptavidin (1:10000). HRP was detected by adding TMB substrates and the reactions were terminated with 1.0 M $H_3PO_4$. For IgG determination, the plates were incubated with peroxidase-conjugated rabbit anti-mouse IgG1 (1:10000) or IgG2a (1:10000) and developed by the addition of ABTS solution. Then optical density was measured on a Sunrise Absorbance Reader (TECAN, Austria) at 450 nm (for TMB substrate) and 415 nm (for ABTS substrate).

## Cytokine measurements

Splenocytes of mice were cultured in the 24-well plates at a density of $1 \times 10^6$ cells/ml and sub-sequently stimulated with or without Per a 2. After stimulation for 3–5 days, protein levels in cell culture supernatants were implemented via the Bio-Plex multiplex immunoassay (mouse 23-plex panel; Bio-Rad Laboratories Inc., USA) to detect IL-1α, IL-1β, IL-2, IL-3, IL-4, IL-5, IL-6, IL-9, IL-10, IL-10, IL-13, IL-17A, Eotaxin, G-CSF, GM-CSF, IFN-γ, MCP-1, MIP-1α, MIP-1β, RANTES, and TNF-α. As described in the manufacturer's protocol, protein concen-trations were measured based on a standard curve.

## Quantitative real-time PCR

The extraction of total RNAs from the skin lesions was conducted and the Per a 2-stimulated splenocytes for 3 days using TRIzol reagent (Thermo Fisher Scientific, MA, USA), based on the manufacturer's recommendation. cDNA synthesis was prepared with a SuperScript III kit (Invitrogen CA, USA) using isolated RNAs as templates and oligo $(dT)_{15}$ as a primer. Quanti-tative PCR was performed to analyze the expression of cytokines on the StepOnePlus system (Applied Biosystems, CA, USA). The predesigned primer sequences are listed in S1 Table. The data of gene expression were imported into an Excel database and represented as a fold increase between untreated cells and allergen-stimulated cells following normalization with the housekeeping gene β-actin.

## Batch fermentation of recombinant Per a 2 *L. lactis* strain for oral administration

The recombinant *L. lactis* strain NZ3900 harboring pNZ8149-Per a 2 grown in M17 broth was cultured using a bench-top fermenter (Firstek, Taiwan). Fermentation was carried out at 30˚C and the pH was adjusted to 6.8–7.2 with a 2N NaOH solution. Recombinant Per a 2 protein was induced with 50 ng/ml nisin for 3 hours, as conducted by pre-experiments [37]. The harvested cell pellets were used for immunoblotting to quantify the expressed Per a 2 protein using purified *E*-rPer a 2 as standards.

## Immunoblotting for quantification of expressed Per a 2 protein in recombinant *L. lactis*

Harvested cells of Per a 2 *L. lactis* were s dispersed in PBS and disturbed by sonication using a Branson digital sonifier (Branson Ultrasonics, Danbury, CT) for 30 minutes on ice. Samples of cell lysates were separated on a 4–12% polyacrylamide gel using Laemmli's method. For immunoblotting, the gel was moved to a nitrocellulose membrane and following probed with rabbit anti-rPer a 2 antibody. Subsequent detection was performed using a peroxidase-labeled goat anti-rabbit IgG antibody (10000-fold dilution, Millipore) and 3-amino-9-ethyl carbazole (AEC) as a substrate of enzyme.

## Statistical analysis

Data analysis was conducted with SPSS software (version 20, SPSS Inc., Chicago, IL) using the Dunnett *t*-test or the Bonferroni multiple range test. *P*-values less than 0.05 were indicated as statistical significance.

# Results

## Adjuvant-free sensitization to aeroallergen Per a 2 through skin exposure

We demonstrated previously that mice can be sensitized to cockroach allergen, Per a 2 adsorbed to alum adjuvant through intraperitoneal injection [37, 40]. Here, we tested the hypothesis that aeroallergens initiate sensitization in the skin. Mice were exposed to purified *E*-rPer a 2 by painting it on the abdominal skin once weekly for 5 exposures without adjuvant. Serum levels of total IgE (Fig 3A), Per a 2-specific IgE (Fig 3B), and IgG1 (Fig 3C) antibody response were increased significantly, but not Per a 2-specific IgG2a (Fig 3D).

## Skin histopathology and scratch behavior response to aeroallergen Per a 2 following intradermal challenge

Upon intradermal injection of *E*-rPer a 2, evidence of the immediate-allergic reaction was determined by examining scratching in sensitized mice. Significantly increased numbers of scratching bouts were found only in the Per a 2 group-50 compared with the control mice ($p<0.05$) (Fig 4). Compared with the scratching background of the PBS group (means ± SD = 42 ± 19), the counts in the sensitized groups of Per a 2–10 (73 ± 33) and Per a 2–100 (103 ± 45) also showed an increase, though without statistical significance.

Fig 5A depicts the pathologic changes in the skin from each group 48 hours post-Per a 2 challenge by H&E staining. The aeroallergen-painted mice exhibited significantly increased numbers of eosinophils, neutrophils, and lymphocytes in the dermis in a dose-dependent manner, in comparison with the PBS mice (Fig 5B).

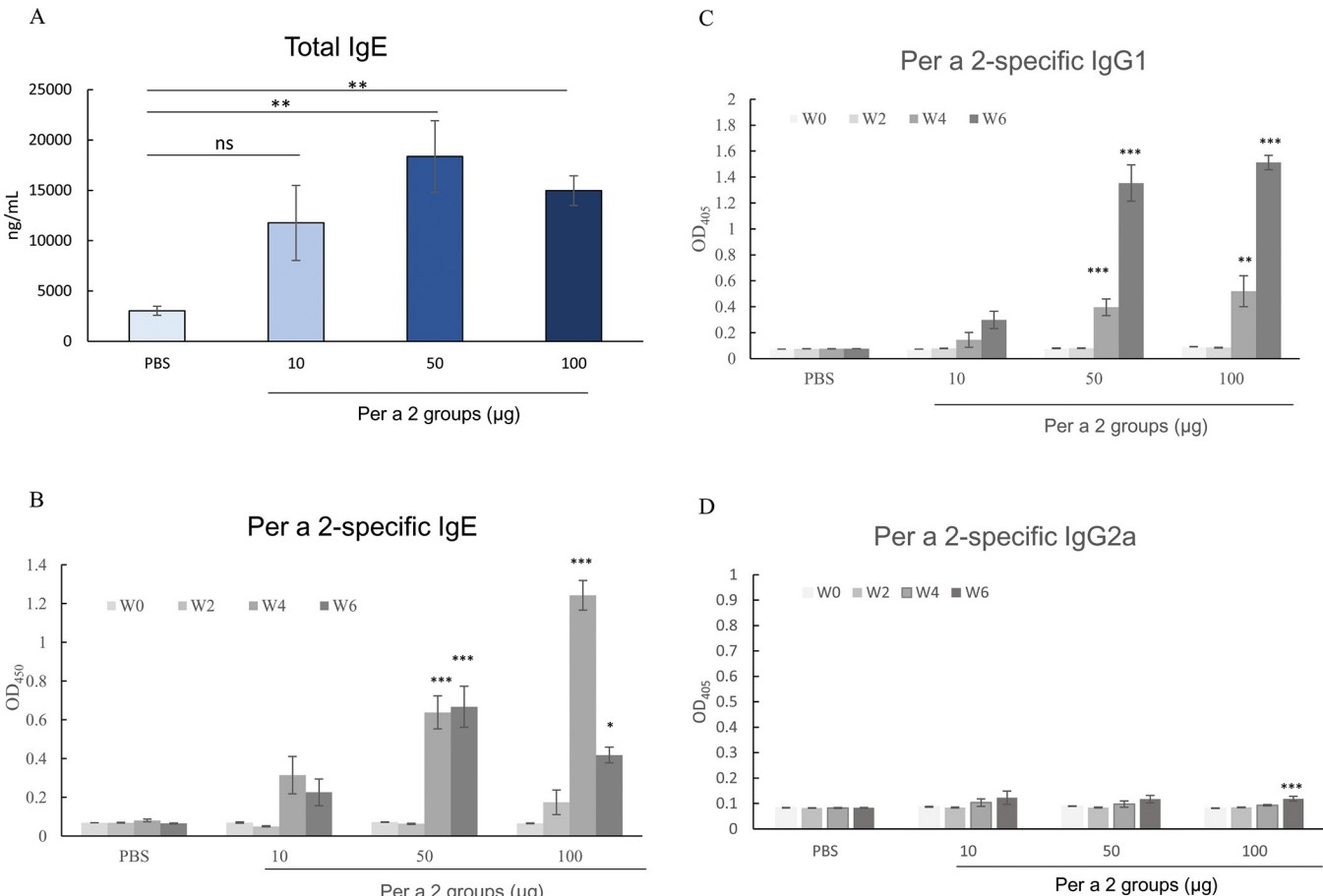

**Fig 3.** Changes in serum total IgE (A) and Per a 2-specific IgE (B), -IgG1(C), and -IgG2a (D) antibodies in murine groups as indicated in weeks. * denotes *p*<0.05, ** denotes *p*<0.01, *** denotes *p*<0.001, ns denotes not statistically significant by one-way analysis of variance with the Bonferroni multiple range test.

### Cytokine production by *E*-rPer a 2-stimulated splenocytes

To confirm that aeroallergen Per a 2 could induce allergic inflammation, the cytokine profiles of *E*-rPer a 2-stimulated splenocytes were analyzed, as shown in Fig 6. Levels of Th2 cytokines, IL-4, IL-5, IL-9, and IL-13, were significantly induced in a dose-dependent manner (Fig 6A), but not T helper 1 (Th1)-type cytokines (Fig 6B). The results were further verified by examining the aeroallergen-induced mRNA expression levels of *IL-4*, *IL-13*, and *IFN-γ* using real-time PCR in *E*-rPer a 2-stimulated splenocytes (Fig 6C). The results supported the above findings that showed aeroallergens induced a significant increase in Th2 cytokine RNA levels of splenocytes, but not *IFN-γ*.

Taken together, our study revealed that the optimal dose of *E*-rPer a 2 was 50 μg/painting. Sensitization was therefore performed with this dose in the therapeutic experiments of *L. lactis*-Per a 2.

### Oral administration of *L. lactis*-Per a 2 ameliorated Per a 2-induced scratch behaviors and specific antibody responses

For AD patients, the primary goal of treatment is to provide relief from the itchy skin. *L. lactis*-Per a 2, either at 1- or 2-μg doses, significantly ameliorated allergen-induced scratch bouts in

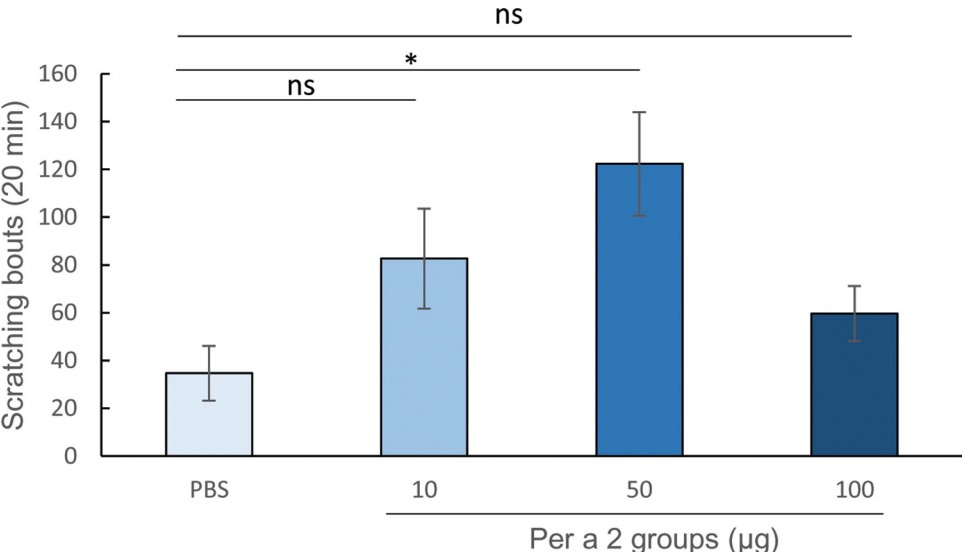

**Fig 4. Scratching bouts of mice from each group.** The scratching counts were performed for 20 minutes after induction by intradermal injection of rPer a 2 at the endpoints of the experiment. The statistical significance of differences between the PBS group and Per a 2 groups was analyzed by Dunnett's test. * denotes $p<0.05$, ns denotes not statistically significant.

sensitized mice, as shown in Fig 7A. Serum total IgE from the T2-Per a 2 group was significantly reduced (Fig 7B). Moreover, both therapeutic groups had significantly lower levels of Per a 2-specific IgE (Fig 7C) and IgG1 (Fig 7D) in comparison with the control sham group.

## Oral administration of *L. lactis*-Per a 2 suppressed the inflammation of skin lesions and splenic *IL-4* and *IL-13* mRNA expression in Per a 2-induced AD mice

Treatment with *L. lactis*-Per a 2 significantly attenuated Per a 2-induced dermatitis, as the numbers of total infiltrating cells were decreased compared to the T-sham group (Fig 8A). The thickness of the dermis was also reduced in both treatment groups. The mRNA expressions of *TSLP* and *IL-31*, cytokines involved in AD pathogenesis, in the skin lesions of the therapeutic groups were also significantly decreased (Fig 8B). Moreover, the splenic mRNA expressions of *IL-4* and *IL-13* were also significantly downregulated in the therapeutic groups compared to the T-sham group (Fig 8C).

## Discussion

IgE-mediated sensitization to airborne allergens has been found to be related to AD [18]. Allergens exist in a variety of environments encountered in daily life. Previously, we revealed a substantial presence of Per a 2 protein within roach feces and this protein exhibited heightened resistance to degradation in the environment, which therefore produces a much higher concentration, resulting in sensitization [20]. Epithelial barrier dysfunction, one of major risk factor for AD, may facilitate the epicutaneous penetration of allergens, leading to the development of epicutaneous sensitization [9]. The sensitization and challenge protocol were modified from Kitamura's protocol, in which 2,4-dinitrofluorobenzene (DNFB) was painted on the dorsal skin of the mice [41]. In the present study, our protocol was a total of 5 exposures without adjuvant through repeated painting of Per a 2 onto the skin. Consistent with

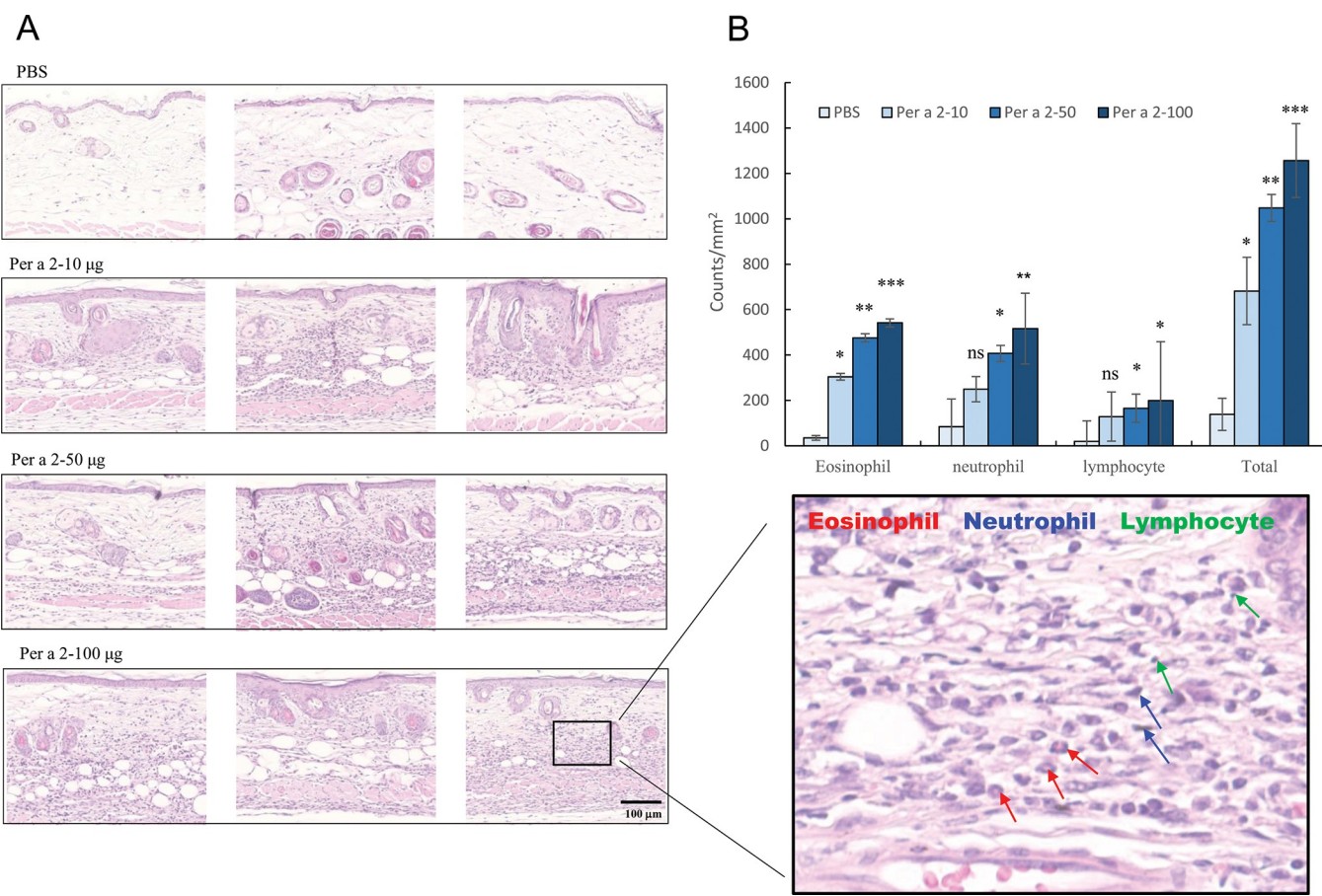

**Fig 5. Histopathology and cell infiltration in skin lesions from Per a 2-sensitized mice.** (A) Histopathology of Per a 2-challenged skins from each group by H&E staining. Figures indicated the skin sections obtained 48 hours after intradermal challenge under 400 x light microscope. (B) The infiltrating inflammatory cells were recorded. Data are expressed as the mean±SEM (N = 5). Data were analyzed using one-way analysis of variance followed by Dunnett's test. * denotes $p<0.05$, ** denotes $p<0.01$, *** denotes $p<0.001$, and ns denotes not statistically significant versus the control (PBS) group.

previously reported results, we found that repeated brushing of Per a 2 without adjuvant onto the skin induced significant epidermal thickening, significantly more scratch bouts, as well as significantly increased cell counts of eosinophil, neutrophil, and lymphocyte in a dose-dependent manner. Our study results provide good evidence supporting the epicutaneous sensitization route of allergens, which resembles the natural course of human AD.

Consequently, an AD murine model was established and used for further evaluation of the therapeutic effects of oral administration of Per a 2 *L. lactis*. Our results showed that oral feeding of *L. lactis*-Per a 2 ameliorated Per a 2-induced scratch behaviors and decreased the production of total IgE, Per a 2-specific IgE, and IgG1. Previous studies also reported consistent findings, suggesting that serum IgE, total IgG, and IgG1, but not IgG2a antibody levels, were significantly greater in a midge-sensitized mouse model [38, 39]. The treatment of *L. lactis*-Per a 2 also suppressed inflammatory infiltration, as well as mRNA expressions of *TSLP* and *IL-31* in skin lesions, and downregulated splenic *IL-4* and *IL-13* transcription in Per a 2-induced AD mice. The roles of IL-4, IL-13, IL-31, and TSLP in the pathogenesis of atopic dermatitis might involve a dysfunctional epidermal barrier and allergen sensitization, which are characterized by intense itching and pruritus [4, 10, 18]. IL-33 enhances the secretion of pruritic cytokines, including TSLP and IL-31, from keratinocytes and Th2 cells, respectively. IL-31 was

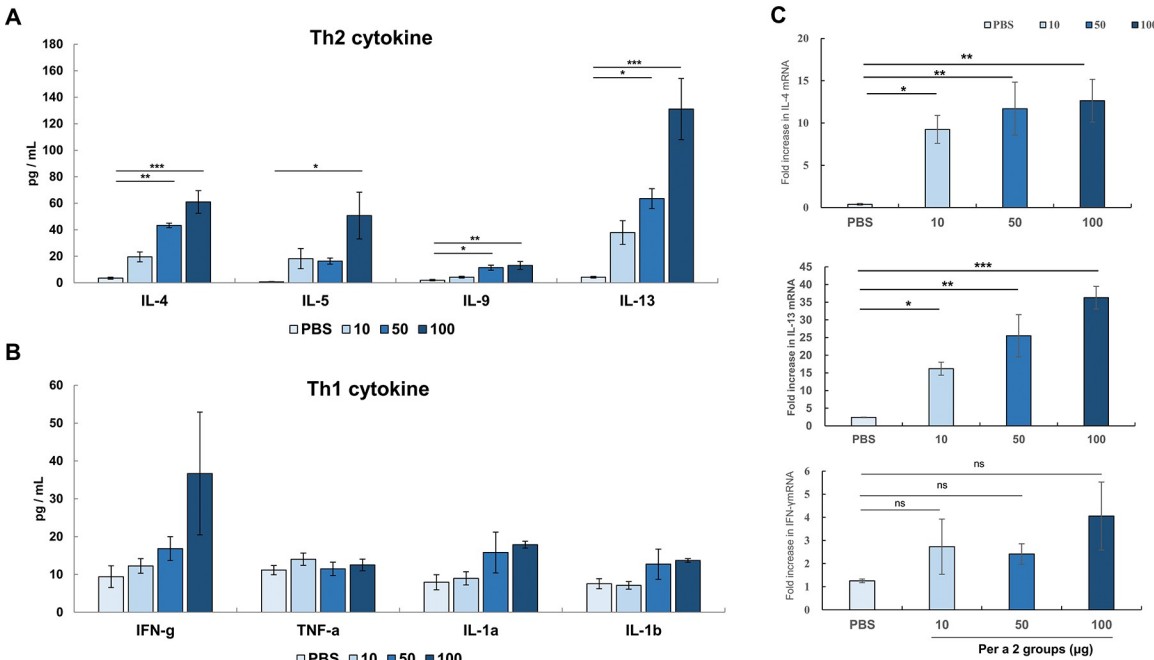

**Fig 6. Th2 and Th1 immune response of Per 2-stimulated splenocytes in mice from each group.** Protein levels of Th2 (A) and Th1 (B) are measured by multiplex immunoassay. (C) mRNA expression levels of IL-4, IL-13, and IFN-γ by real-time PCR. Data are expressed as mean ± SEM. The statistical significance of differences between groups was analyzed by Dunnett's test. * denotes $p<0.05$, ** denotes $p<0.01$, *** denotes $p<0.001$, ns denotes not statistically significant.

strengthened in lesion and non-lesion skin of AD patients and IL-31 directly inhibits the differentiation of keratinocytes, which results in the disturbance of epidermal barrier function through itch-induced scratching [42, 43]. Our results support the notion that these cytokines might have potential as targets for oral tolerance of AD therapies.

Allergen avoidance is the best method, theoretically, for preventing allergic disease and manifestation. However, Per a 2 and Bla g 2, the major group 2 German cockroach allergens, could exist for more than a year in the household environment [20, 44]. It might be difficult for allergic patients to avoid these long-term cockroach allergens.

Currently, there are treatments for AD, but many have drawbacks. ASIT via subcutaneous injection requires weekly/bi-weekly injection and some adverse effects have been reported in many randomized controlled trials [31–33]. Subcutaneous immunotherapy (SCIT) increased local adverse effects by 25% and systemic adverse effects by 5% [31]. Sublingual immunotherapy (SLIT) increased local adverse effects by 5% and systemic adverse reactions by 0.05%, respectively [31]. *L. lactis* has been widely used as a delivery vector for antigenic and therapeutic proteins via mucosal routes of administration [35]. Upon oral administration, the *L. lactis* reach the intestine and are taken up by the microfold (M) cells of Peyer's patches where they are delivered across the epithelium to the underlying antigen-presenting cells to elicit specific immune responses [35, 36]. Finally, *L. lactis* might release the recombinant protein to induce immunological tolerance. Previously, we developed a triple-aeroallergen vaccine in pNZ8149/*L. lactis* NZ3900 against the top three major indoor aeroallergens, Der p 2 (mite), Per a 2 (roach), and Cla c 14 (mold), and proved its efficacy and safety for prevention of respiratory allergy in a mouse model [37]. Consistent with the present study, we have demonstrated the allergen-specific therapeutic effects of *L. lactis*-derived Per a 2 in Per a 2-painted AD model. Our data indicated that a total of only 25 doses of oral Per a 2 containing *L. lactis* at 1–2 μg

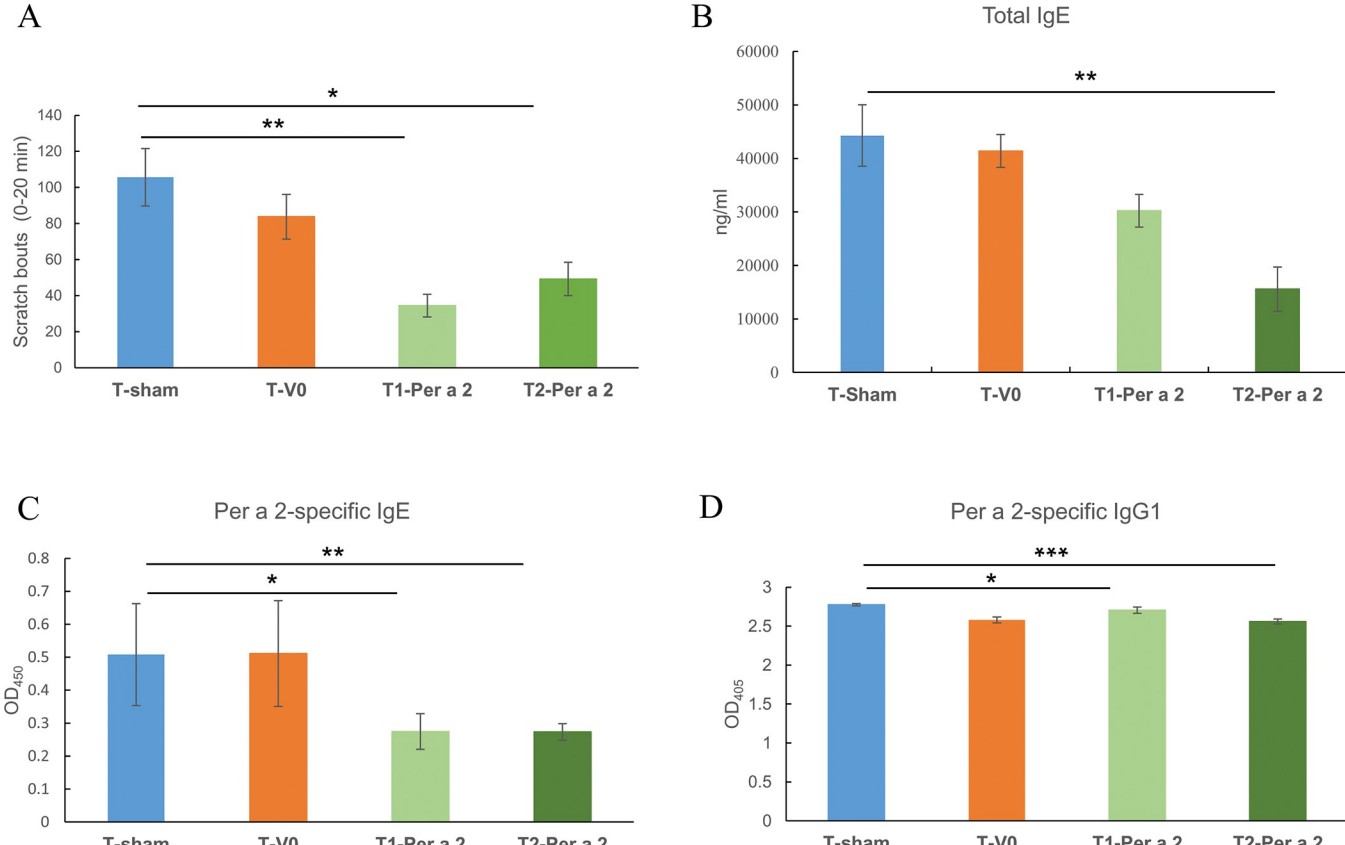

**Fig 7. Effects of *L. lactis*-Per a 2 oral treatment on scratch behaviors and antibody responses in AD mice.** (A) Scratching bouts, (B) Total IgE, (C) Per a 2-specific IgE, and (D) -IgG1. Data are expressed as mean ± SEM. The statistical significance of differences was analyzed by Dunnett's test. * denotes $p < 0.05$, ** denotes $p < 0.01$, *** denotes $p < 0.001$ as compared with the AD control group (T-sham).

doses resulted in a significant decrease in type 2 inflammation of AD compared with the control mice. Our findings indicate that OT could exert its effects through the induction of allergen-specific blocking antibodies and regulatory T cells to achieve immunological tolerance and could potentially treat the root cause of the allergic disease and stop the progression of atopic march.

Several limitations of this study must be considered. First, since AD is a multifactorial inflammatory skin disease, there remains a translational gap between AD-like mouse models and human AD [45]. Second, although American cockroach is the second most common aeroallergen, environmental allergens, such as house dust mites, pollens, and animal epithelia allergens could simultaneously contribute to AD [4, 10]. In future investigations, we plan to develop a pronged multi-allergen-specific treatment for AD. Third, as the scope of this investigation was restricted to the therapeutic effects of oral allergen-specific tolerance, in further studies, we also aim to determine whether early intervention with oral administration of *L. lactis*- Per a 2 would limit the progression of atopic march in children with AD.

In conclusion, our study provides evidence showing that repeated brushing of cockroach allergen on the skin leads to atopic dermatitis and that a therapeutic strategy involving cockroach allergen-specific oral tolerance may be feasible. These findings may open new avenues of research into the treatment of atopic dermatitis and preventing the progression of atopic march.

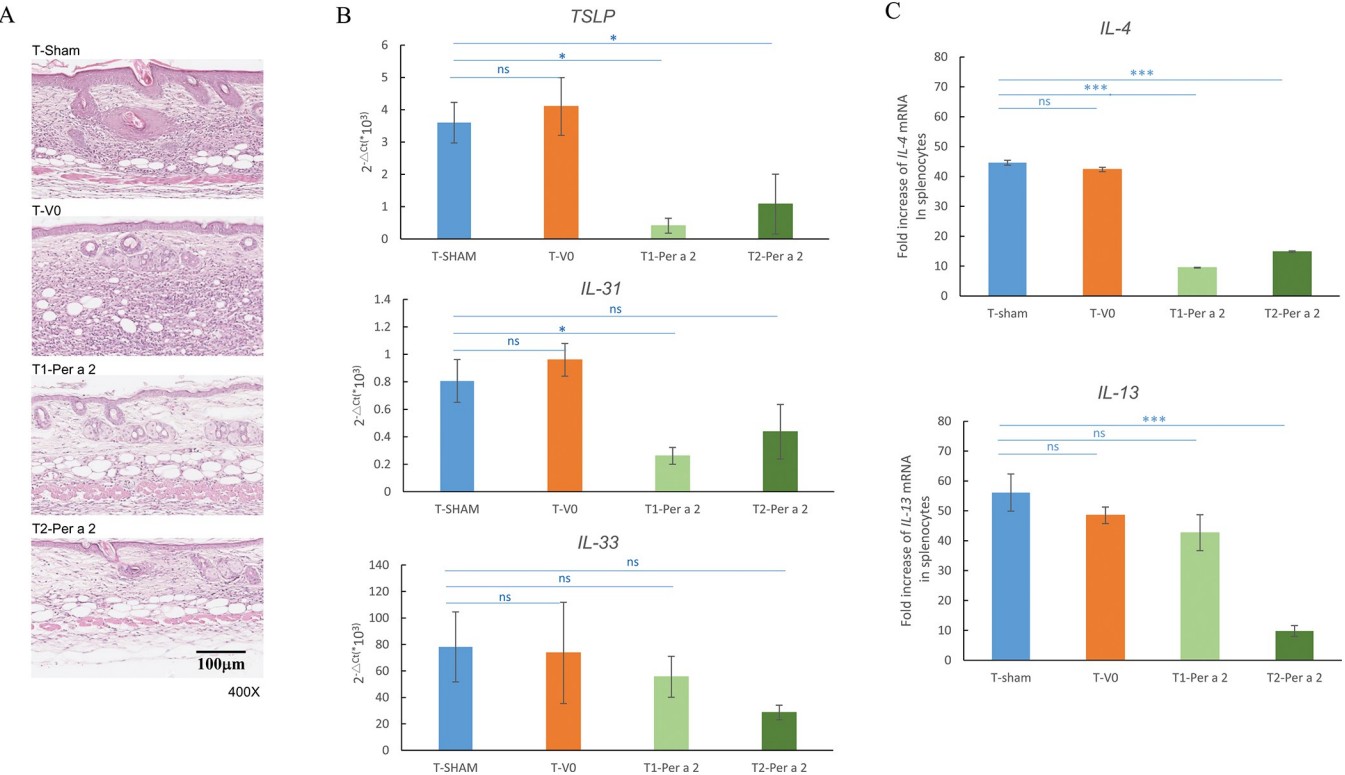

**Fig 8. Effect of *L. lactis*-Per a 2 oral treatment in skin lesions of sensitized mice.** (A) H&E staining of sections and (B) real-time PCR data for cytokines TSLP, IL-31, and IL-33 in skin lesions or (C) IL-4 and IL-13 in splenocytes from the four groups. The statistical significance of differences between sham and therapeutic groups was analyzed by Dunnett's test. Data are expressed as the mean ±SEM of 6 mice. * $p < 0.05$, ns denotes not statistically significant.

## Supporting information

**S1 Table. The sequences of murine gene-specific primers used in real-time PCR.**
(DOCX)

## Author Contributions

**Conceptualization:** Mey-Fann Lee, Yi-Hsing Chen, Yu-Wen Chu.

**Data curation:** Chi-Sheng Wu.

**Formal analysis:** Mey-Fann Lee, Chi-Sheng Wu.

**Investigation:** Mey-Fann Lee, Ming-Hao Lee.

**Methodology:** Mey-Fann Lee, Chi-Sheng Wu, Ming-Hao Lee.

**Project administration:** Yu-Wen Chu.

**Supervision:** Yu-Wen Chu.

**Validation:** Mey-Fann Lee, Nancy M. Wang, Yi-Hsing Chen.

**Writing – original draft:** Mey-Fann Lee, Yu-Wen Chu.

**Writing – review & editing:** Nancy M. Wang, Yu-Wen Chu.

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
