## [Decision Letter · Decision Letter 0]

13 Jun 2023

PONE-D-23-04327An atopic dermatitis-like murine model by skin-brushed cockroach Per a 2 and oral tolerance induction by Lactococcus lactis-derived Per a 2PLOS ONE

Dear Dr. Chu,

Thank you for submitting your manuscript to PLOS ONE. After careful consideration, we feel that it has merit but does not fully meet PLOS ONE’s publication criteria as it currently stands. Therefore, we invite you to submit a revised version of the manuscript that addresses the points raised during the review process.

We look forward to receiving your revised manuscript.

Kind regards,

Hossam M Ashour, Ph.D.

Academic Editor

PLOS ONE

- https://bmcvetres.biomedcentral.com/articles/10.1186/s12917-015-0514-6#citeas

In your revision ensure you cite all your sources (including your own works), and quote or rephrase any duplicated text outside the methods section. Further consideration is dependent on these concerns being addressed.

3. To comply with PLOS ONE submissions requirements, in your Methods section, please provide additional information regarding the experiments involving animals and ensure you have included details on (1) methods of sacrifice, (2) methods of anesthesia and/or analgesia, and (3) efforts to alleviate suffering.

5. Please upload a new copy of Figures 2,4,5,6 and 7 as the detail is not clear. Please follow the link for more information: https://blogs.plos.org/plos/2019/06/looking-good-tips-for-creating-your-plos-figures-graphics/" https://blogs.plos.org/plos/2019/06/looking-good-tips-for-creating-your-plos-figures-graphics/

Additional Editor Comments:

All comments need to be addressed.

Reviewers' comments:

Reviewer's Responses to Questions

**Comments to the Author**

1. Is the manuscript technically sound, and do the data support the conclusions?

Reviewer #1: Yes

Reviewer #2: Yes

Reviewer #3: Yes

2. Has the statistical analysis been performed appropriately and rigorously? 

Reviewer #1: Yes

Reviewer #2: Yes

Reviewer #3: Yes

3. Have the authors made all data underlying the findings in their manuscript fully available?

Reviewer #1: Yes

Reviewer #2: Yes

Reviewer #3: Yes

4. Is the manuscript presented in an intelligible fashion and written in standard English?

Reviewer #1: No

Reviewer #2: Yes

Reviewer #3: Yes

5. Review Comments to the Author

Reviewer #1: 1. The entire manuscript must be proofread and extensively verified for grammatical errors by an English specialist.

2. Use Italics for the representation of gene names wherever applicable.

3. Spell out the word when used first time in the whole manuscript (for eg TSLP, Th1, etc)

Results:

1. All figure legends should be written either with the figure or separately (not in the results).

2. The author should also include the unit of protein concentration in the figures.

3. How does the author justify the rise in per-a 2-specific IgG1 but not IgG2a? In figure 2,

mice were exposed to purified E-rPer a 2. Does the author examine the concentration of

other IgGs?

4. In figure 3, if the author doesn't have statistically significant data, how can they say that

Groups Per a 2-10 and 100 has higher scratching counts?

5. In fig 4, why the author did check only three inflammatory cell (lymphocytes, neutrophils

and eosinophils). Please explain. What is the method by which they checked the level of

these inflammatory markers?

6. In the discussion part, the author should describe the role of IL-33, IL-31 and TSLP in the

pathogenesis.

Reviewer #2: The manuscript entitled "An atopic dermatitis-like murine model by skin-brushed cockroach Per a 2 and oral tolerance induction by Lactococcus lactis-derived Per a 2 " is well described. The authors should add a paragraph to indicate limitations of their study. The significance of the findings need to be discussed in more detail. Titles for the figures should be added.

Reviewer #3: The represented manuscript “An atopic dermatitis-like murine model by skin-brushed cockroach Per a 2 and oral tolerance induction by Lactococcus lactis-derived Per a 2” is an interesting study. However, it lacks some important characterization. My comments are the following.

1. Authors should consider putting the vector design for recombinant proteins in the method section.

2. The introduction is lacking the current AITs like SCIT and SLIT. Please discuss the importance of OIT as well.

3. The Oral administration of Lactis bacterial strain is not clear. Please detail it in the methodology section.

4. The discussion is poorly written, lacking the importance of OIT in respective of others ASIT approaches (i.e., SCIT, SLIT). Also, the importance of IgG subtypes is not discussed well in context of model development and treatment.

5. Authors are claiming the mechanism of AD sensitization and treatment. However, it's hard to find any mechanism in the manuscript. Please try to discuss the possible mechanism of skin-based sensitization and oral treatment. What's good in skin to create sensitization? Secondly how does Lactis strain induce oral tolerance? Is it secreting protein or dyeing to release the recombinant protein? and where these things are happening in body stomach, or intestine, or mouth or some other places?

6. The study is missing data of lung tissues (i.e., cell infiltration in lungs, cytokine analysis, mucus deposition etc.)

7. St Table is missing GAPDH primers sequences.

8. How much abdominal area has been painted to get sensitization. Please specify it in the method section.

9. The counting of different cells in histosections is not clear. Can you please put the arrow on the cell type in histosections? For me, everything looks similar. Please also include the scale bar in images.

10. Fig.6 D, the statistics are confusing. Kindly recheck it.

11. Again, histoimages are not making good sense here. Please try to be explanatory in the images. What are they showing?

12. The timeline of oral ASIT is missing. Kindly put in the figure before serum data.

6. PLOS authors have the option to publish the peer review history of their article (what does this mean?). If published, this will include your full peer review and any attached files.

Reviewer #1: No

Reviewer #2: No

Reviewer #3: **Yes: **Akhilesh Kumar Shakya, Ph.D., Texas Tech University

---

## [Author Response · Author response to Decision Letter 0]

9 Aug 2023

Reviewer #1:

1. The entire manuscript must be proofread and extensively verified for grammatical errors by an English specialist.

Reply: The main text has been sent for English editing by a qualified native English-speaking expert, as suggested. The editorial certificate will be uploaded with the revised manuscript. 

2. Use Italics for the representation of gene names wherever applicable.

Reply: Thank you for pointing this out. We have used italics for all gene names in the revised manuscript. 

3. Spell out the word when used first time in the whole manuscript (for eg TSLP, Th1, etc)

Reply: Thank you for this comment. We have revised the whole manuscript accordingly.

Results:

1. All figure legends should be written either with the figure or separately (not in the results).

Reply: We have incorporated an additional paragraph of figure legends in the final section on Page 24.

2. The author should also include the unit of protein concentration in the figures.

Reply: Thank you for this suggestion. We have added the protein concentrations for the Per a 2 groups in the figures.

3. How does the author justify the rise in per-a 2-specific IgG1 but not IgG2a? In figure 2, mice were exposed to purified E-rPer a 2. Does the author examine the concentration of other IgGs?

Reply: In general, allergen-specific IgE plays a primary role in immediate hypersensitivity reactions, and it is true that other isotypes of IgG, such as IgG1 can also induce similar reactions in mouse models (Lehrer et al. Immunology 32 (4):507-511, 1977; Xiang et al. Int. Arch. Allergy Immunol. 141 (2):168-171, 2006; Lee et al. PLOS ONE 9(3): e91871, 2014). As shown in Figure 2C, sensitization in mice showed a marked increase in Per a 2-specific IgG1 (but not IgG2a, Figure 2D) in a dose- and time-dependent manner by ELISA.

4. In figure 3, if the author doesn't have statistically significant data, how can they say that Groups Per a 2-10 and 100 has higher scratching counts?

Reply: Thank you for noticing this and we apologize for the confusion. To make this clearer, the sentence has been revised as follows “...Compared with the scratching background of the PBS group (means ± SD = 42 ± 19), the counts in the sensitized groups of Per a 2-10 (73 ± 33) and Per a 2-100 (103 ± 45) also showed an increase, though without statistical significance.” (Lines 228-230)

5. In fig 4, why the author did check only three inflammatory cell (lymphocytes, neutrophils and eosinophils). Please explain. What is the method by which they checked the level of these inflammatory markers?

Reply: In AD, skin barrier disruption followed by a proinflammatory cytokine milieu and inflammation play an important pathogenic role. Specific inflammation markers can be detected by blood biochemistry/ELISA or flow cytometry. However, in order to realistically represent and obtain multiple biopsy samples of the inflammatory environment including various cell types, environmental cues, and cellular structure, the biopsied mice AD specimens were assessed by histopathological examination in this study. Histopathological analysis of skin lesions usually includes quantification of white blood cells to evaluate the inflammatory phase. Eosinophil infiltration of the skin is a common finding in hypersensitivity reactions and also represents an important cellular link between innate and adaptive immune responses. By releasing cytokines, eosinophils have the potential to attract neutrophils and T lymphocytes at sites of inflammation. The methods of counting different cells have been added on page 7-8. “As described previously, the quantification of infiltrating inflammatory cells of each mouse were conducted within H&E-stained skin sections by medical technicians [38].” (Lines 143-145)

6. In the discussion part, the author should describe the role of IL-33, IL-31 and TSLP in the pathogenesis.

Reply: We have added a description regarding the role of these three cytokines in the discussion section (Lines 290-297).

 

Reviewer #2: The manuscript entitled "An atopic dermatitis-like murine model by skin-brushed cockroach Per a 2 and oral tolerance induction by Lactococcus lactis-derived Per a 2 " is well described. The authors should add a paragraph to indicate limitations of their study. The significance of the findings need to be discussed in more detail. Titles for the figures should be added.

Reply: We thank the reviewer for this helpful suggestion. The limitations of this study and the significance of the findings have been addressed in the revised discussion section (Lines 321-329 and Lines 317-320), as suggested. We have incorporated an additional paragraph of figure legends with titles in the final section on Page 24-25.

Reviewer #3: The represented manuscript “An atopic dermatitis-like murine model by skin-brushed cockroach Per a 2 and oral tolerance induction by Lactococcus lactis-derived Per a 2” is an interesting study. However, it lacks some important characterization. My comments are the following.

1. Authors should consider putting the vector design for recombinant proteins in the method section.

Reply: The following sentences have been added to lines 97-99: “As previously described expression of American cockroach allergen Per a 2 recombinant proteins were used vectors pET30 in Escherichia coli [22]. Plasmid pET30 contains a His-tag sequence, which is expressed as a stretch of six histidine residues at the N-terminal end of the target protein…….”

2. The introduction is lacking the current AITs like SCIT and SLIT. Please discuss the importance of OIT as well.

Reply: We thank the reviewer for this helpful suggestion. We have added some information regarding ASIT and the importance of OT in the introduction section (Lines 76-81 and 90-93).

3. The Oral administration of Lactis bacterial strain is not clear. Please detail it in the methodology section.

Reply: The oral administration methodology has been added to lines 127-131, as follows: “The sensitization and treatment schedule is summarized in Table 1. Mice were intra-gastrically (IG) administered 200 μl of syrup containing 1 μg (as group T1-Per a 2) or 2 μg (T2-Per a 2) of the expressed Per a 2 protein once a day on weekdays for a total of 25 times during days 7 to 40. Control mice (group T-VO) received E-rPer a 2 at the same time of painting and oral administration of NZ3900 L.lactis alone.”

4. The discussion is poorly written, lacking the importance of OIT in respective of others ASIT approaches (i.e., SCIT, SLIT). Also, the importance of IgG subtypes is not discussed well in context of model development and treatment.

Reply: We apologize for the confusion. The description has been revised accordingly on Page 15 in Lines 302-306 and Page 16 in Lines 317-320. The importance of IgG subtypes has been addressed on Page 14, in Lines 286-288. 

5. Authors are claiming the mechanism of AD sensitization and treatment. However, it's hard to find any mechanism in the manuscript. Please try to discuss the possible mechanism of skin-based sensitization and oral treatment. What's good in skin to create sensitization? Secondly how does Lactis strain induce oral tolerance? Is it secreting protein or dyeing to release the recombinant protein? and where these things are happening in body stomach, or intestine, or mouth or some other places?

Reply: We thank the reviewer for this helpful suggestion. Atopic dermatitis often precedes the development of other atopic diseases, allergic rhinitis, and allergic asthma (Davidson et al. J Allergy Clin Immunol 143(3): 894-913, 2019). Therefore, we developed the epicutaneous sensitization model to study the efficacy of oral tolerance induction by Lactococcus lactis-derived Per a 2. The mechanism of oral tolerance has been addressed in the revised discussion section (Lines 308-311).

6. The study is missing data of lung tissues (i.e., cell infiltration in lungs, cytokine analysis, mucus deposition etc.)

Reply: Our previous studies indicated that cockroach allergen Per a 2 could induce respiratory allergies and is correlated with asthma severity (Lee et al. Ann Allergy Asthma Immunol 108: 243-248, 2012). However, AD is generally the first allergic manifestation of atopic march, which is then followed by other allergic disorders, such as allergic rhinitis, asthma, and food allergy. In this study, we aimed to develop an early intervention with oral tolerance of L. lactis-Per a 2 that would stop the atopic march in children with AD. 

7. St Table is missing GAPDH primers sequences.

Reply: The housekeeping gene for PCR normalization used in this study is β-actin, as described in line 188.

8. How much abdominal area has been painted to get sensitization. Please specify it in the method section.

Reply: The following sentence has been added to lines 118-120: “The size of the area on the shaved abdominal skin that was painted with sensitization solution measured about 1 cm2.

9. The counting of different cells in histosections is not clear. Can you please put the arrow on the cell type in histosections? For me, everything looks similar. Please also include the scale bar in images.

Reply: We apologize for the confusion. To make this clearer, we have revised Figure 4A to include arrows in different colors to denote different cell types, and the revised Figure 7A now has a scale bar.

10. Fig.6 D, the statistics are confusing. Kindly recheck it.

Reply: We apologize for the confusion. The sentence on Page 12 in lines 254-255 has been revised, as follows: “Moreover, both therapeutic groups had significantly lower levels of Per a 2-specific IgE (Fig 6C) and IgG1 (Fig 6D) comparing with the control sham group.”

11. Again, histoimages are not making good sense here. Please try to be explanatory in the images. What are they showing?

Reply: We have uploaded new tiff files for all figures to improve the quality of images.

12. The timeline of oral ASIT is missing. Kindly put in the figure before serum data.

Reply: Thank you for your comment. 

We have incorporated a timeline scheme of oral ASIT in Table 1.

---

## [Decision Letter · Decision Letter 1]

24 Aug 2023

An atopic dermatitis-like murine model by skin-brushed cockroach Per a 2 and oral tolerance induction by Lactococcus lactis-derived Per a 2

PONE-D-23-04327R1

Dear Dr. Chu,

We’re pleased to inform you that your manuscript has been judged scientifically suitable for publication and will be formally accepted for publication once it meets all outstanding technical requirements.

Kind regards,

Hossam M Ashour, Ph.D.

Academic Editor

PLOS ONE

Additional Editor Comments (optional):

Reviewers' comments:

Reviewer's Responses to Questions

**Comments to the Author**

1. If the authors have adequately addressed your comments raised in a previous round of review and you feel that this manuscript is now acceptable for publication, you may indicate that here to bypass the “Comments to the Author” section, enter your conflict of interest statement in the “Confidential to Editor” section, and submit your "Accept" recommendation.

Reviewer #1: All comments have been addressed

Reviewer #2: All comments have been addressed

Reviewer #3: All comments have been addressed

2. Is the manuscript technically sound, and do the data support the conclusions?

Reviewer #1: Yes

Reviewer #2: Yes

Reviewer #3: Yes

3. Has the statistical analysis been performed appropriately and rigorously? 

Reviewer #1: Yes

Reviewer #2: Yes

Reviewer #3: Yes

4. Have the authors made all data underlying the findings in their manuscript fully available?

Reviewer #1: Yes

Reviewer #2: Yes

Reviewer #3: Yes

5. Is the manuscript presented in an intelligible fashion and written in standard English?

Reviewer #1: Yes

Reviewer #2: Yes

Reviewer #3: Yes

6. Review Comments to the Author

Reviewer #1: Dear Editor,

The investigative group of the manuscript “An atopic dermatitis-like murine model by skin-brushed cockroach Per a 2 and oral tolerance induction by Lactococcus lactis-derived Per a 2” have satisfactorily answered all queries and now ready to publish.

Thanks

Reviewer #2: The manuscript can be accepted in its current form. The authors have addressed all comments raised before.

Reviewer #3: Authors revision is satisfactory. They have taken care maximum reviewers comments. I think, editor may consider it for publication.

7. PLOS authors have the option to publish the peer review history of their article (what does this mean?). If published, this will include your full peer review and any attached files.

Reviewer #1: No

Reviewer #2: No

Reviewer #3: **Yes: **Akhilesh Kumar Shakya

---

## [Editor Report · Acceptance letter]

29 Aug 2023

PONE-D-23-04327R1 

An atopic dermatitis-like murine model by skin-brushed cockroach Per a 2 and oral tolerance induction by *Lactococcus lactis*-derived Per a 2 

Dear Dr. Chu:

I'm pleased to inform you that your manuscript has been deemed suitable for publication in PLOS ONE. Congratulations! Your manuscript is now with our production department. 

Kind regards, 

on behalf of

Dr. Hossam M Ashour 

Academic Editor

PLOS ONE